# Food as Culture among African Women: Exploring Differences between North and South (Morocco-Senegal)

**DOI:** 10.3390/foods11162433

**Published:** 2022-08-12

**Authors:** E. Begoña García-Navarro, María José Cáceres-Titos, Miriam Araujo-Hernández

**Affiliations:** 1Nursing Department, University of Huelva, 21007 Huelva, Spain; 2Contemporary Thinking and Innovation for Social Development Research Center (COIDESO), Social Studies and Social Intervention Research Center (ESEIS), Faculty of Nursing, University of Huelva, 21007 Huelva, Spain

**Keywords:** culture, eating habits, health promotion, cultural identity, nourishment

## Abstract

The goal of this exploratory study was to analyze the influence of culture on African women’s diet considering their role as primary caregivers. The analysis differentiated between Moroccan and Senegalese women and identified the key elements that influence their dietary habits and their health. Using a qualitative methodology, we performed a triangulation of data based on a literature review and a panel of experts, all of which served as the basis for the interview script to conduct 14 semi-structured interviews (n = 7 Moroccan and n = 7 Senegalese). This study reflects the substantial relationship between dietary habits, cultural identity, and health that healthcare providers need to acknowledge. It is important for healthcare practitioners to be culturally competent in order to provide holistic and individualized care.

## 1. Introduction

Nourishment is a basic need for human evolution, which has undergone important transformations. Food has gone from being understood as a means of subsistence to a matter influenced by biological, geographic, psychological, and cultural factors that determine how we behave, how we live, and how we relate to others [1]. Studies focusing on this topic have prioritized the positivist paradigm, emphasizing the nutrient itself rather than specific eating behaviors; this reflects the “prominence of caloric values over symbolic values” [2] without taking into account the relationship between cultural practices and eating habits, which are closely related to overall health among the population, making their study and examination crucial [3]. The scientific community has found that the complexity of nutritional issues goes beyond aspects that are merely physical, physiological, and quantifiable. This highlights the key role of social and cultural issues, which, after all, are the determining factors in the construction of eating-related ideologies and representations of what is and what is not healthy food [4,5]. Diet is not solely a medical prescription with a healing purpose; the main feature of eating behaviors involves the social practices that surround them. These practices are embedded in the culture, customs, traditions, policies, norms, and values of each social group, which are influenced by geography, time, economic models, and policies, particularly health policies [6].

It is noteworthy that the influence of culture is rooted in historical traditions that tend to legitimize certain eating and drinking social practices. For instance, in France, wine has long been associated with healthy habits [7]. The epistemic function enables a conceptual framework for sensemaking, while the affiliation function can convey a cultural baggage that consolidates the social representations of food in a given culture, binding tradition and eating/drinking habits while connecting food and cultural identity [4,5,6].

In this context, it is important to recognize the direct impact of eating habits on health, whereby the concept of health must be added to the diet–cultural identity pairing, creating a triad that should not be overlooked by healthcare providers. Another study describes how cultural identity can modify people’s interpretation of nutritional information and, consequently, their food consumption habits [8]. Health-conscious patients or consumers capable of taking action for their life goals and preferences will be consistent with food and health as long as their culture integrates them into their habits [9]. Therefore, personal nutrition researchers and providers should devise strategies to inform and discuss with patients or consumers the implications of their life goals and health preferences.

Although there are many studies that discuss existing relationships between particular foods and certain diseases, this study aims to understand, through the testimonies of the women themselves, their perspectives and perceptions regarding the factors that condition health in relation to food as well as the elements that influence it, highlighting the importance of culture in decision making.

## 2. Materials and Methods

### Study Design

A qualitative exploratory study with a phenomenological approach was conducted by means of a content analysis as described by Taylor and Bodgan [10]. This research adheres to the COREQ guidelines [11].

In order to address the proposed research objectives, a data triangulation [12] was carried out, comprising an initial bibliographic search that, together with the panel of experts, served as the basis for the interview script. These resulting categories were validated using a “modified Delphi” methodology, where, based on the opinions of the experts, a consensus was reached, which made it possible to obtain the interview guide to be used with the participants [13]. The sampling strategy for the expert panel was based on theoretical sampling with a total of 7 professional experts from different disciplines (Table 1).

To conclude the methodological triad, we began with semi-structured interviews targeted at two populations. On the one hand, we selected Moroccan women living in Spain or Morocco, and on the other hand, we selected Senegalese women living in Spain or Senegal. Given the difficulties of accessibility to recruit volunteers as informants, a snowball sampling technique was used [14], reaching 7 Moroccan women and 7 Senegalese women, achieving saturation in each of the dimensions [15].

The semi-structured interviews were conducted through open-ended questions, allowing the participants to argue and elaborate their answers, lasting between one and one and a half hours during the months of December 2020 and June 2021. Fieldwork continued until no new data could be extracted in relation to the thematic categories.

The inclusion criteria required female participants to be over 18 years of age, to live or have lived in an area of Morocco or Senegal (depending on their group), to participate voluntarily, and to have read and signed the informed consent form.

For the analysis of the interviews, the model described by Taylor-Bodgan [10] (Table 2) was used.

The research team listened to recordings and read interview transcripts to make an initial cursory interpretation. This provided a general idea that supported a deeper analysis (this involved the identification of relevant recurring themes, the search for similarities and differences between themes to develop codes—dimensions—and, with these, thematic categories). The presence of coinciding codes—dimensions—by different researchers—blind analysis—indicated that the analysis reached the core and revealed the meanings of the phenomenon under study.

In order to ensure validity and reliability, the entire process of coding and discourse analysis was carried out independently by three members of the research team. Discrepancies were discussed until a consensus was reached.

The study was conducted in accordance with the Declaration of Helsinki guidelines [16].

## 3. Results

### Characteristics of the Participants

The present study was conducted with 14 women, namely 7 Moroccan women (MW) and 7 Senegalese women (SM), who were representative in terms of age, geographic area, religion, level of education, and interview language, with some participating women not sharing our mother tongue (Table 3).

Coding of discourse was carried out using the interview transcripts, and it produced 11 different codes. Data were analyzed via discourse analysis using ATLAS.ti 9 Scientific Software Development GmbH, Berlin, Germany. Considering repetition patterns in expressions in relation to each code, four lines of argumentation emerged. These were evenly distributed within both population groups, suggesting the determining factors influencing the perception of food coincide in both populations.

Through observation, expert interviews, and interviews with women, different categories emerged underlying those initially raised.

Such themes arose after reflection on how the study population thinks and comments on their own health and well-being.

Through their discourses, we learned about these women’s perspectives and perceptions of the factors that condition health in relation to food as well as the elements that influence them; this enabled us to move to the interpretation stage, increasing the level of abstraction.

Their health self-concept is a unique aspect, which is evident in the discourse by participants from both populations. In this context, there are different reflections on the importance of maintaining a healthy diet in order to achieve good health and how nutrition conditions their health (Table 4).

There are multiple elements that have an impact on eating habits and, consequently, on their health. Among them, the resources and materials are noteworthy as well as their religion, the impact of culture, resource accessibility, and health promotion investment and practices. These factors contribute to the consumption of certain types of food and limit access to others, also affecting the way they are cooked and ingested (Table 4).

**Religion** invites activities that promote health, such as healthy eating, hygiene, and prayer, which are very present in both populations. These constitute key aspects in their perception of health (Figure 1).

The respondents stated that their diet follows patterns dictated by their religion. For example, Muslim women say that their religion prevents them from eating pork or that they prepare certain foods in a particular way. This religion has a symbiotic relationship with culture, and their mutual influence generates some remarkable eating habits (Table 5).

**Culture** is expressed through rituals and traditions and becomes yet another element that influences the foods that are eaten and when and why. With participants from both Muslim and Christian countries, we found interesting differences between them, such as, for example, the Feast of the Lamb or Ramadan for Muslims and avoiding meat or fasting on Fridays during Lent for Christians (Table 5).

On the other hand, a common and very frequent diet staple in both populations is tea. Their culture assigns a significance to tea that goes beyond its nutritional or hydration values, adding a cultural and customary value.

**Accessibility** to food resources conditions the type of food consumed as well as the perception of its value for their health self-concept.

In their discourses, a multifactorial self-concept of health was revealed, which conditions the **health promotion** initiatives that they develop and assume as communities and as individuals (Table 5).

Regarding the elements that generate **health differences** (Table 6) among the Senegalese and Moroccan populations, two important subcategories were observed: **gender** and **economic resources** (Figure 2).

Gender conditions the distribution, responsibilities, and modes of participation in family and cultural nutrition as well as specific elements of the female condition. Women are the main caregivers responsible for feeding their entire family, while tea and its cultural significance is the responsibility of men.

Pregnancy involves a special process and a stage of life that highlights the importance of taking care of nutrition.

One of the emerging categories found in this study is infertility, associated with current eating habits, which are considered poorer than in the past.

Socioeconomic level is a key element that all the participants, both Moroccans and Senegalese, mentioned as a decisive factor that influences their diet and, ultimately, their health. People with fewer resources buy cheaper and less diverse foods as opposed to those with a higher economic level.

Regarding lifestyle habits and gender, there are two significant findings in relation to sports and physical care; they explain that it is more common to see men engaging in some kind of physical activity than women.

The consequences on the population’s health (Table 6) were reflected in two emerging categories: on the one hand, the concern among members of the Moroccan population for cancer and, on the other, its connection with lifestyle and dietary habits.

The interviewees reflected that food goes beyond fulfilling biological needs but has become a social value and a cultural ritual of solidarity and companionship throughout history. In Senegal, the value they place on eating together as a family is particularly significant and is considered a tradition in itself.

Finally, participants highlighted how the global pandemic of Covid19 (Table 6) conditioned their dietary practices. Due to the subsistence economy, many people were unable to work, so food was also affected, and they began to change eating patterns and consume what was available to them.

## 4. Discussion

It is important to understand the role of eating behaviors among humans and how they have evolved in order to understand why current eating habits exist and how they influence health.

Our study has shown how people’s eating habits are influenced by factors that are not exclusively biological, such as culture, social class, age, education, health, and even their social environment [17]. Globalization and the scientific and technological revolution have led to the recognition of the complexity of the food process and the need for analysis from a holistic perspective that integrates biological principles, social practices, and the motivations that structure subjects’ eating processes [18].

The results of our study demonstrate that habits do not only satisfy physiological and psychological needs but also respond to cultural needs and gender needs, among other needs. This applies because people tend to show their social status, their economic level (especially if it is high), and their values. However, the meanings attributed to this status are not inherent to food since they depend on the social context in which they are developed. Food thus has both a material and a symbolic meaning. These results are consistent with several authors [19] who assert that distinct social groups have different habits and behaviors, which specifically present the needs of each group, their cultural conditions, their experiences, and their history, among other aspects, thus establishing a particular distinction that can be recognized. These authors also agree with another finding from our research, which refers to the way people feed themselves as a reflection of their social identity and their belonging to a particular social group [19,20,21].

The cultural importance of food and nutrition is centered on the values, meanings, and social beliefs that are present in today’s society.

When choosing food, people are influenced by a series of internal aspects, such as biological and emotional factors, beliefs, and attitudes. There are also social, economic, and physical elements that also influence food choices; these are external factors [22]. Several studies [23,24] establish that there is a statistically significant relationship between eating patterns and the aforementioned variables. In our study, participants’ statements evidence the presence of these variables in dietary decision making, also taking into account gender and religion.

A study conducted at Dalhousie University, Canada [25], examined the meaning of food for Goan women in Toronto and the role of food in creating and maintaining gender-differentiated ethnic identities. The overlaps with our results lie in how the gender role of women in caring for and cooking food had a particular power or “currency” within the family and community, valued for fostering and supporting cultural identity; another study developed in the same country, at the University of Alberta [26], matched the food, culture, and gender triad as part of the ethnic identity within the community, as can be seen in some of our informants’ discourses. In this case, the respondents, the study population of Moroccan women residing in Canada, agreed, concluding that the women shared their struggles to maintain ethnic cuisine as a marker of community affiliation that is related to preservation of health in the community despite the difficulties to access food, whether financial or due to the lack of resources in the country itself.

A study developed at the University of Oviedo, Spain, with the African population [27] supports our main findings along the dimensions of religion, culture, and gender as determinants of health in the food sphere, and it also coincides with the religious prescriptions and eating rituals without taking into account the consequences for health although they affirm that health begins with adequate eating.

## 5. Conclusions

African women’s understanding of food-related health is influenced by culture, gender, and the identity of the community where they reside.

Religion, both in North African and Sub-Saharan African women (both practice Islam although in Senegal, it is associated with Islam-Animism), has a great influence on the definition of food discourse, and it is remarkable how Islam defines what can be eaten or not, what is allowed by the Muslim community, and what is associated with different religious rituals. There is a radical ban on any food derived from pork, which is a ban derived from religious prescriptions.

The preservation of their own cultural dietary patterns is especially favored by the family, where women play a fundamental role as agents of health in their own community.

African women’s conception of health in the study is conditioned by dietary prescriptions linked to their culture. These are also associated with the deprivation of banned substances such as alcohol and tobacco although most of the women interviewed linked this not only to culture but also to economic resources, which are fundamental for good health.

## Figures and Tables

**Figure 1 foods-11-02433-f001:**
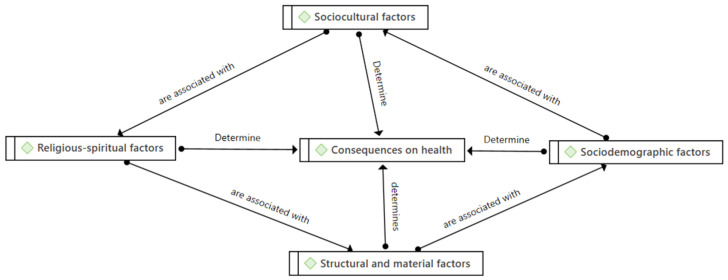
Health conditioning factors. Source: Prepared by the authors.

**Figure 2 foods-11-02433-f002:**
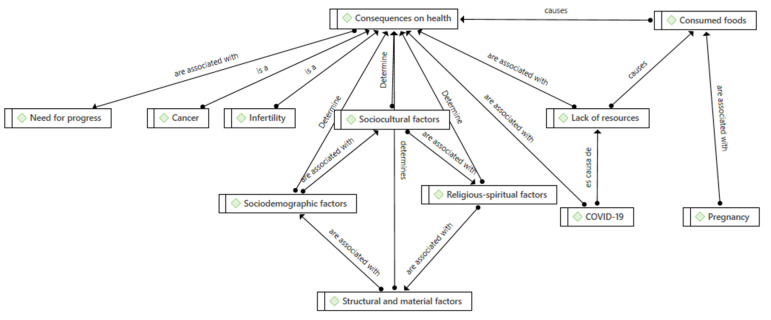
Gender-related health consequences and lack of resources. Source: Prepared by the authors.

**Table 1 foods-11-02433-t001:** Characteristics addressed in the expert panel.

Categories/Experts	EX1	EX2	EX3	EX4	EX5	EX6	EX7
Gender	Female	Male	Female	Female	Male	Male	Female
Age	59	47	32	39	45	37	44
Profession	Nurse	Doctor	Social worker	Dietitian	Nurse	Dietitian	Doctor
Health self-concept	“The concept of health is related to illness, which changes between cultures.”	“Eating a varied diet, with fruit and vegetables, is one of the essential factors that makes them feel healthy.”	“Each person’s health is individual and is influenced by their beliefs.”	“Health care is related to dietary care, influenced by the norms dictated by their religion.”	“Health is conditioned by culture, religion, and even place of residence, as accessibility to resources is not the same for a person in an urban area as it is for a person in a rural area.”	“Because of their culture and religiosity, although health is good and important, it does not come first. It is more important to have a job or to feed themselves.”	“They tend to have a more positivist concept of health, not so much focused on holistic health.”
Health differences	“Economic status influences the resources they can access.”	“Gender is a key factor that causes many differences in health.”	“Although it is true that having few economic resources has a negative impact on health, there is a very positive feeling of mutual support.”	“It is necessary to talk about women, people living in rural areas and having few economic resources as possible elements that have a negative impact on health.”	“Men are generally healthier than women.”	“A person’s purchasing power favours his or her health by having access to more resources.”	“It is not as easy for a man as it is for a woman to achieve optimal health because of the different roles they play.”
Consequences on health	“In many cases health is measured in the absence of disease.”	“More and more cases of cancer are associated with healthy lifestyles and habits.”	“The strong belief in a superior being sometimes makes them not to take responsibility for the illness, leaving it “to the will of God”.”	“The aforementioned differences in health have a direct impact on health and can lead to numerous diseases.”	“Eating a varied diet based on natural products is a protective factor against the disease.”	“There is fear of serious diseases, but not enough health promotion to prevent them.”	“There is no health-centred care, but there is no desire to get sick either. It is difficult to work on these two aspects.”
COVID-19	“COVID has had a negative impact on the lives of all people.”	“At first there was fear, but the subsistence economy has prioritised starvation over COVID.”	“Having to stop working has sometimes meant that many people did not even have enough to eat.”	“Many people have been afraid and very exposed. There has been a lack of resources to deal with this situation.”	“Religiosity has undoubtedly influenced the process of coping with this situation.”	“Given the choice between dying of Covid19 or starving their family to death, most would prefer the former.”	“All the elements mentioned when talking about health consequences can be applied when talking about COVID.”

**Table 2 foods-11-02433-t002:** Analytical process in qualitative research (Taylor-Bogdan).

Stage	Task
Discovery (search for topics by browsing the data in various ways)	Read the data repeatedly Follow up on themes, ideas, and interpretations Search for emerging themes Create topologies Develop theoretical concepts and propositions Read the literature Develop a story guide
Coding (collection and analysis of all data relating to themes, ideas, concepts, interpretations, and propositions)	Develop coding categories Code the dataset Categorize the data according to coding scheme Check which data remain Refine the analysis
Data revitalization (interpreting data in the context where it was collected)	Data requested or not requested Influence of the observer on the context Who was there? (Differences between what people say and do when they are alone and when others are there) Direct and indirect data Source (distinction between the perspective of a single person and that of a larger group) Our own assumptions (critical self-reflection)

**Table 3 foods-11-02433-t003:** Participants’ sociodemographic characteristics.

Participant	Age	Place of Origin	Place of Residence	Religion	Educational Level	Language of the Interview
MS1	37	Dakar	Huelva	Muslim	Until 17 years of age	Spanish
MS2	26	Fatick	Fatick	Muslim	High School Diploma	French
MS3	32	Dakar	Dakar	Christian	Until 17 years of age	French
MS4	43	Ziguinchor	Dakar	Christian	High School Diploma	French
MS5	26	Dakar	Huelva	Christian	High School Diploma	Spanish
MS6	26	Baker	Dakar	Muslim	High School Diploma	French
MS7	27	Niakhar	Niakhar	Muslim	No education	Serere-French
MM1	38	Oujda	Huelva	Muslim	Until 17 years of age	Spanish
MM2	24	Agadir	Huelva	Muslim	Until 17 years of age	Spanish
MM3	22	Rabat	Huelva	Muslim	Until 17 years of age	Spanish
MM4	40	Oujda	Huelva	Muslim	Until 17 years of age	French
MM5	38	Rabat	Huelva	Muslim	High School Diploma	Spanish
MM6	22	Fkih ben Salah	Huelva	Muslim	Associate Degree	Spanish
MM7	21	Larbaat al aounat	Salé	Muslim	High School Diploma	French

**Table 4 foods-11-02433-t004:** Origin of categories and subcategories by population group.

Dimension	Subcategory	Moroccan Women	Senegalese Women
Health self-concept	Resources and materials	Yes	Yes
	Religion	Yes	Yes
	Culture	Yes	Yes
	Accessibility	Yes	Yes
	Health promotion	Yes	Yes
Health differences	Gender	Yes	Yes
	Economic level	Yes	Yes
Consequences on health	Coping with illness	Yes	Yes
	Solidarity	No	Yes
	Cancer	Yes	Yes
COVID-19		Yes	Yes

**Table 5 foods-11-02433-t005:** Health self-concept: Senegalese and Moroccan women’s statements.

Dimension	Population	Quote	Sub-category	Population	Quote
Health self-concept	MS3	“For me to be healthy is to have good health, good hygiene, a healthy life, eat good food, do sports if possible, although we do not have this culture, I have discovered it here in Spain. Even if it costs money, health is priceless.”	Resources and materials	MS7	“I live in an area where it is very difficult to access fish and there are many families who don’t even consume it because they don’t have the money to buy it.”
			Religion	MM7	“Culture is something you share in society, the type of clothes you wear, for example. Religion tells you that you have to be healthy and that you have to observe the Ramadan, but culture is what tells you how to prepare tea or food.”
			Culture	MM1	“If they offer you tea and you turn it down, it may even be impolite.”
				MM4	“The culture in each place is the food that is eaten, how and why it is made, the dance performed, and so on.”
	MM2	“What I like most about traveling to Morocco is being able to buy fruit and vegetables there. In Spain, for example, potatoes don’t have any flavor, and I think that’s because you use a lot of chemicals.”		MS1	“It’s usually the man of the family who prepares the tea and is in charge of serving it.”
				MS4	“In Senegal, tea is a ritual specific to the teranga (Senegalese hospitality), where it is consumed as a community.”
				MM2	“In our religion, the Muslim religion, many holidays are associated with food, such as Ramadan or the Feast of the Lamb. The Feast of the Lamb is very important and it is very expensive. People sell their things, to buy the lamb because for Moroccan culture it is a shame if you don’t do it. The religion says, if you don’t have money, people will give you half of theirs so no one will go hungry.”
				MS3	“ We Christians, unlike Moroccans, can eat meat and we don’t observe Ramadan, although we have other traditions.”
	MM7	“I think I am in good health because I take care of my diet. I eat fruits and vegetables.”	Accessibility	MS7	“I live in an area where it is very difficult to access fish and there are many families who don’t even take it because they don’t have the money to buy it.”
			Health promotion	MM5	“There is still a long way to go in education and health. They take care of their food and use natural products. There are some who don’t pay attention to their health, just like everywhere else in the world. Others do what is good for their health like eating healthy.”

**Table 6 foods-11-02433-t006:** Differences and consequences on health and the COVID-19 pandemic: Senegalese and Moroccan women’s statements.

Dimension	Sub-Category	Population	Quote
Differences in health	Gender	MS2	“Pregnant women attend prenatal visits and try to take better care of their diet.”
		MM4	“It’s getting harder and harder to have children, maybe because of the food. It has taken me 5 years to have children.”
		MM7	“Infertility is not discussed, they don’t usually have this problem because they don’t have the infertility-enhancing elements such as late age or food.”
		MS3	“If you walk along the beach, you can see some men doing sports, our culture is healthier because there is no alcohol or tobacco consumption.”
	Economic level	MS3	“Women who have money take better care of their bodies, but for example, I don’t have time. I am at home and I have to take care of the children, the food, the house... it is difficult for me to take care of my health.”
		MM6	“If women are more liberal and have more money, they take better care of themselves. These [aspects] have a lot of influence. Those who have money get check-ups, do sports, take care of their diet, get vaccinated.”
Consequences on health	Solidarity	MS7	“Our family always eats together, all from the same big plate. If someone wants to come and eat with us, they are invited.”
	Cancer	MM5	“There is a lot of cancer. Lately everyone hears that so-and-so has cancer. I think it’s a matter of diet, stress...”
COVID-19		MM2	“People at the beginning did use the mask, but now they only do it to avoid fines, they don’t take it seriously. Maybe because they believe in God’s will and because they need to work to be able to feed themselves and live.”
		MS7	“Many families had to stop working and that’s what brings in money to eat, live and raise our children. I don’t think we’re prepared for something like this, I’m really scared.”

## Data Availability

Data are contained within the article.

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
