# Peer review of "Food as Culture among African Women: Exploring Differences between North and South (Morocco-Senegal)"

_foods, 2022, doi:10.3390/foods11162433_

Round 1

Reviewer 1 Report

This is a manuscript of good scientific significance, but it needs to clarify some early related views. I basically agree with the authors that in today’s society, culture, education, habits and other factors have different effects on food patterns. However, it should be noted that in such research, not only the subjects are the research objects, but also the experimenters' own preconceived views also participate in the research. I think the authors should appropriately introduce the views formed by the author group in previous research or before this research, and appropriately discuss the correlations between these views and the newly generated views. This makes it easy to think about the research results of this manuscript comprehensively.

Author Response

We appreciate your interest and opinions about our research article. It is very pleasing to realize this topic is interesting and useful for you. We would like to express our gratitude for reviewing this manuscript and for the proper corrections, which help us conduct more rigorous and quality research.

 We have detailed below the changes that have been made in the original article: “Food as culture among African women. Exploring differences between North and South (Morocco-Senegal)”.

Modifications have been made following the reviewers’ comments. Each modification is indicated by quoting the reviewer´s comment in addition to the page and line number, followed by the change that has been made in response to that comment.

REVIEWER 1

This is a manuscript of good scientific significance, but it needs to clarify some early related views. I basically agree with the authors that in today’s society, culture, education, habits and other factors have different effects on food patterns. However, it should be noted that in such research, not only the subjects are the research objects, but also the experimenters' own preconceived views also participate in the research. I think the authors should appropriately introduce the views formed by the author group in previous research or before this research, and appropriately discuss the correlations between these views and the newly generated views. This makes it easy to think about the research results of this manuscript comprehensively.

First of all, we would like to thank you for taking the time to review this article, as well as for your comments, which undoubtedly help us to improve.

In accordance with the objective of our study, we are totally in agreement with the corrections you have given us and we think thatt, with the improvements made in relationship with the panel of experts as suggested to us (which can be seen from line 69 onwards of the manuscript and in table 1), a clearer and more comprehensive view of the issue is obtained. In this way, based on the bibliographic search carried out, the panel of experts, whose testimonies allow us to reach a consensus on their opinions, and with them, the interview guide to be carried out with the participants in the study was drawn up. We thank them once again for their ideas.

Reviewer 2 Report

This study aimed to understand, through the testimonies of the women themselves, their perspectives and perceptions regarding the factors that condition health in relation to food, as well as the elements that influence it, highlighting the importance of culture in decision making. The design of the study was correct, unfortunately, and my main concern is that only 14 surveys were carried out; 7 Moroccan women (MW) and 7 Senegalese women (SM) were interviewed to evaluate their perspectives and perceptions about the factors that condition health in relation to food, as well as the elements that influence the decision making. Although the authors claimed that participants were representative in terms of age, geographic area, religion, level of education, and interview language; 14 women are not enough to evaluate perspectives and perceptions from a population of 18.5 and 8.5 million Morocco and Senegalese females in 2020, respectively.

If the authors are willing to publish this paper, the data should be statistically representative of the population, in terms of the place of origin or the place of residence because this would influence their perspectives and perceptions regarding the factors that condition health in relation to food. Another minor comment, there are some Spanish words throughout the manuscript.

Author Response

We appreciate your interest and opinions about our research article. It is very pleasing to realize this topic is interesting and useful for you. We would like to express our gratitude for reviewing this manuscript and for the proper corrections, which help us conduct more rigorous and quality research.

 We have detailed below the changes that have been made in the original article: “Food as culture among African women. Exploring differences between North and South (Morocco-Senegal)”.

Modifications have been made following the reviewers’ comments. Each modification is indicated by quoting the reviewer´s comment in addition to the page and line number, followed by the change that has been made in response to that comment.

REVIEWER 2

This study aimed to understand, through the testimonies of the women themselves, their perspectives and perceptions regarding the factors that condition health in relation to food, as well as the elements that influence it, highlighting the importance of culture in decision making. The design of the study was correct, unfortunately, and my main concern is that only 14 surveys were carried out; 7 Moroccan women (MW) and 7 Senegalese women (SM) were interviewed to evaluate their perspectives and perceptions about the factors that condition health in relation to food, as well as the elements that influence the decision making. Although the authors claimed that participants were representative in terms of age, geographic area, religion, level of education, and interview language; 14 women are not enough to evaluate perspectives and perceptions from a population of 18.5 and 8.5 million Morocco and Senegalese females in 2020, respectively.

If the authors are willing to publish this paper, the data should be statistically representative of the population, in terms of the place of origin or the place of residence because this would influence their perspectives and perceptions regarding the factors that condition health in relation to food. Another minor comment, there are some Spanish words throughout the manuscript.

First of all thank you very much for your kind words regarding our study, as well as for its theme.

The appreciation that you refer to regarding the concern for only 14 surveys is very important, but really by using a qualitative and phenomenological paradigm, it subtracts in terms of the "n" of the quantitative and positivist paradigm. Therefore, surveys under this paradigm are not the most appropriate for data collection, but in-depth interviews. According to the theories of the qualitative paradigm, (Taylor, S.J.; Bogdan, R. Introducción a los Métodos Cualitativos de Investigación (Vol. 1); Paidós: Barcelona, España, 1987.; Gerring, John. Case study research: Principles and practices. Cambridge university press, 2006. Goertz, Gary. "Multimethod Research, Causal Mechanisms, and Selecting Cases: The Research Triad." (2017). Levy-Strauss, Jack. "Qualitative methods and cross-method dialogue in political science." Comparative Political Studies 40.2 (2007): 196-214) el número de personas entrevistadas responde al principio de saturación del discurso ( Fusch, P.I.; Ness, L.R. ¿Ya estamos allí? Saturación de datos en investigación cualitativa. 2015, 20, 1408–1416 ) y la selección de los mismos respondieron a los criterios de representación  en términos de edad, área geográfica, religión, nivel de educación y cultura, que en este caso así lo fueron entrevistando  a 7 mujeres marroquíes (MW) y 7 mujeres senegalesas (SW), 14 women are not enough to evaluate perspectives and perceptions from a population de varios millones de personas, pero bajo el paradigma cualitativo, lo importante es que estas mujeres entrevistadas saturaron su discurso (Fusch; Ness, 2015) respect a las categorías seleccionadas para el análisis del discurso de las mimas, al igual que ha ocurrido en diferentes estudios relacionados con esta metodología publicados en esta revista:

  1. Conroy, DM; young, j.; Errmann, To.; Phelps, T. Positioning Phytosanitary Food Treatments: Exploring the Role of Business-to-Consumer Stakeholder Literacy as an Information Keeper in New Zealand. Food 2022 , 11 , 2108. https://doi.org/10.3390/foods11142108

This study investigates the health literacy of phytosanitary treatments by B2C stakeholders, and the subsequent positioning marketing narratives as an outcome of such literacy  con solo 12 entrevistas en profundidad.

  1. Mellor, C.; Embling, R.; Neilson, L.; Randall, T.; Wakeham, C.; Lee, M.D.; Wilkinson, L.L. Consumer Knowledge and Acceptance of “Algae” as a Protein Alternative: A UK-Based Qualitative Study. Foods 2022, 11, 1703. https://doi.org/10.3390/foods11121703.

The aim of this qualitative study was to develop a rich and contextualised understanding of consumer beliefs about the use of algae in novel and innovative food products.  Con solo 34 participants para todo United Kindong

  1. Wu, X.S.; Miles, A.; Braakhuis, A. Attitudes towards Freshly Made and Readily Prepared Texture-Modified Foods among Speech-Language Therapists, Dietitians, and Community-Dwelling Older Adults. Foods 2022, 11, 2157. https://doi.org/10.3390/foods11142157

This study explored how currently available TMFs (including Soft & Bite-Sized, Minced & Moist, and puree) are perceived by key stakeholders. The participants were recruited and divided into five focus groups to attend food test sessions. Study recruitment was conducted through convenience sampling,

  1. Mena, B.; Ashman, H.; Dunshea, F.R.; Hutchings, S.; Ha, M.; Warner, R.D. Exploring Meal and Snacking Behaviour of Older Adults in Australia and China. Foods 2020, 9, 426. https://doi.org/10.3390/foods9040426

This research, like ours, compares two populations, the Chinese with the Australian, and does so through focus groups (a technique with less capacity for discourse analysis compared to in-depth interviews according to theorists). The selected population,  Sixteen (13 female, 3 male) Australian meat consumers, who were selected for having active lifestyles and being 65–79 years old, were recruited for two focus groups (n = 8 in each focus group). Twenty-one (17 female, 4 male) Chinese meat consumers with active lifestyles and being 60–81 years old, participated in two focus groups (n = 12 and 9).

  1. Garrido-Fernández, A.; García-Padilla, F.M.; Sánchez-Ramos, J.L.; Gómez-Salgado, J.; Sosa-Cordobés, E. The Family as an Actor in High School Students’ Eating Habits: A Qualitative Research Study. Foods 2020, 9, 419. https://doi.org/10.3390/foods9040419

This research, developed by colleagues from our university, the authors consider to describe family conceptions and difficulties about healthy eating during the school day and to know the proposals towards improving healthy eating habits in their children. The participants were recruited and divided into five focus groups to attend food test sessions. Study recruitment was conducted through convenience sampling,

**Another example published in this journal with qualitative methodology of our research team, uses a sample of a total of 6 in-depth interviews and 2 focus groups.

  1. Araujo-Hernández, M.; García-Navarro, E.B.; Cáceres-Titos, M.J. Dietary Behaviours of University Students during the COVID-19 Pandemic. A Comparative Analysis of Nursing and Engineering Students. Foods 2022, 11, 1715.

We hope that we have been able to meet your needs with this justification, we continue to await your consistent assessments.

Another minor comment, there are a few Spanish words throughout the manuscript.

With regard to the last point, we have reviewed the article again and corrected the Spanish words that were found in both figures. Thank you again for your comments.

Round 2

Reviewer 2 Report

Dear authors,

The comment was not to interview the whole women population (18.5 and 8.5 million Morocco and Senegalese females in 2020, respectively), but I was expecting at least 140 interviewed people (70 Moroccan and 70 Senegalese women). The fact that there are exploratory studies based on qualitative data with less than 14 people in the sample does not mean that is the correct way of doing this type of study. However, your clarification is enough.

Comment to the paper: just specify what were the modifications of the Delphi method.

On the other hand, I suggest revising social-psychological models like the theory of reasoned action (TRA; Ajzen and Fishbein, 1980) and the theory of planned behavior (TPB). These models have been used to explore the rationality that underlies the individual’s decisions to engage in a given behavior and the contribution of factors influencing it (Zubair and Garforth, 2006). The TRA has been used as an analytical framework, which explicitly recognizes the importance of the influence of actions and behavior, as well as the attitudes and perceptions of the decision maker (Garforth et al., 2004).

Suggested references:

1.    Ajzen, I. and Fishbein, M. (1980). Understanding Attitudes and Predicting Social Behaviour. Upper Saddle River, NJ: Prentice-Hall.

2.    Garforth, C., Rehman, T., McKemey, K., Tranter, R., Cooke, R., Yates, C., Park, J. and Dorward, P. (2004). Improving the design of knowledge transfer strategies by understanding farmer attitudes and behaviour. Journal of Farm Management 12:17–32.

Zubair, M. and Garforth, C. (2006). Farm level tree planting in Pakistan: The role of farmers’ perceptions and attitudes. Agroforestry Systems 66:217–229.